# What Has a Foundation Model Found?
# Using Inductive Bias to Probe for World Models

Keyon Vafa [1]  Peter G. Chang [2]  Ashesh Rambachan [2]  Sendhil Mullainathan [2]

## Abstract

Foundation models are premised on the idea that sequence prediction can uncover deeper domain understanding, much like how Kepler's predictions of planetary motion later led to the discovery of Newtonian mechanics. However, evaluating whether these models truly capture deeper structure remains a challenge. We develop a technique for evaluating foundation models that examines how they adapt to synthetic datasets generated from some postulated world model. Our technique measures whether the foundation model's inductive bias aligns with the world model, and so we refer to it as an *inductive bias probe*. Across multiple domains, we find that foundation models can excel at their training tasks yet fail to develop inductive biases towards the underlying world model when adapted to new tasks. We particularly find that foundation models trained on orbital trajectories consistently fail to apply Newtonian mechanics when adapted to new physics tasks. Further analysis reveals that these models behave as if they develop task-specific heuristics that fail to generalize.

## 1. Introduction

The promise of foundation models relies on a central presumption: that learning to predict sequences can uncover deeper truths, or optimistically, even a world model. While this idea is new in one sense, it is old in another. Hundreds of years ago, astronomers like Kepler discovered geometric patterns that could pinpoint the future locations of planets in the night sky. Newton would later expand on this progress to develop Newtonian mechanics, fundamental laws that could not only predict the movement of planets but also explain physical properties across the universe (Koestler, 1959; Gin-

gerich, 2004). This path — from predicting sequences to understanding the deeper mechanisms that underlie them — is not unique to physics. In biology, animal breeders noticed patterns in the traits of offspring long before their predictive insights inspired Mendel to develop a theory of genetics.

How would we know if foundation models have also made the leap from making accurate predictions to developing reliable world models? This paper develops a framework for answering this question. Specifically, we create a procedure that, when given a foundation model and world model, tests whether the foundation model has learned that world model. We call this technique an *inductive bias probe*, and it is built on a simple insight: the implicit world model of a foundation model is revealed by how it extrapolates from a small amount of information. This is inspired by how scientists use world models — to make inferences from small amounts of data. Similarly, the inductive bias of a foundation model reveals its world model.

We first demonstrate this procedure using an example from physics. Specifically, we aim to replicate Kepler's and Newton's experiments, albeit replacing the physicist with a foundation model of orbital mechanics. Much like Kepler, the model is able to predict orbital trajectories, even for solar systems it has not seen.

What would it mean for this foundation model's inductive bias to be toward Newtonian mechanics? We demonstrate one tangible way to test this: we fine-tune the foundation model on a small dataset where the output is exactly the force vector (a cornerstone of Newtonian mechanics) at each point in the trajectory. If the foundation model's world model is toward Newtonian mechanics, it should have an inductive bias towards these force vectors. In contrast, Figure 1 shows that the model produces poor force vectors. More extremely, when we perform this exercise at a larger scale across many solar systems, the laws of gravity it uses to generalize bear no resemblance to Newton's law (Table 1).

We further apply inductive bias probes in other domains with a known world model: lattice problems and Othello games (Liu et al., 2022; Hazineh et al., 2023; Nanda et al., 2023b; Vafa et al., 2024). Across these domains, we find that neural sequence models have weak inductive biases

[1]Harvard University [2]MIT. Correspondence to: Keyon Vafa <kvafa@g.harvard.edu>.

*Proceedings of the $42^{nd}$ International Conference on Machine Learning*, Vancouver, Canada. PMLR 267, 2025. Copyright 2025 by the author(s).

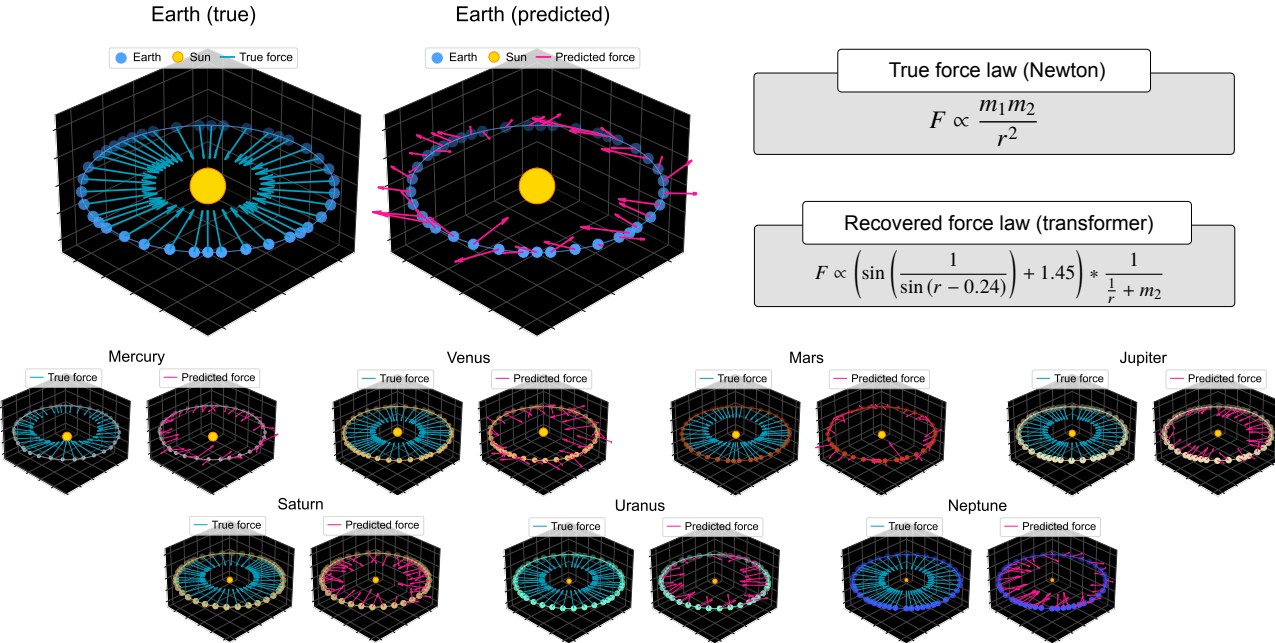

*Figure 1.* Each pair of panels illustrates the trajectory of a planet in the solar system and its gravitational force vectors, comparing the true Newtonian forces (left) to the predicted forces (right) from a transformer foundation model pretrained on orbital sequences and fine-tuned to predict forces. While the model excels at generating accurate predictions of planetary trajectories, it does not have an inductive bias toward true Newtonian mechanics; moreover, its force predictions recover a nonsensical force law, as revealed by symbolic regression.

toward the given world models. We also highlight a practical implication: models that perform better on inductive bias probes have better performance when they're fine-tuned to perform new tasks that rely on the underlying world model.

Taken together, our results provide a direction for understanding the deficiencies of foundation models: if a model's inductive bias isn't toward a known model of reality, what is it toward? We explore this question by examining whether these foundation models have alternative inductive biases. Our analysis reveals that these models instead behave as if they develop task-specific heuristics that fail to generalize. For physics, rather than learning one universal physical law, the foundation model applies different, seemingly nonsensical laws depending on the task it's being applied to. In lattice and Othello, models have an inductive bias toward the set of legal next-tokens (e.g. a board's legal next moves) rather than the world model itself.

## 2. Framework

In this section, we lay out our framework for evaluating whether a foundation model has learned a postulated world model. We develop an *inductive bias probe*, which is a procedure that evaluates the foundation model's behavior as it adapts to new tasks.

**Data and tasks.** Let $x \in \mathcal{X}$ denote an input and $y \in \mathcal{Y}$ de-note some output. A dataset $D = \{(x_1, y_1), \ldots, (x_n, y_n)\}$ is a collection of $n$ input-output pairs. A *task* $f \colon \mathcal{X} \to \mathcal{Y}$ is a mapping between inputs and outputs.

**Foundation models:** A *foundation model* is a learning algorithm which, when given a dataset $D$, returns a prediction function $\widehat{m}_D \colon \mathcal{X} \to \mathcal{Y}$ that relates the input to the outputs. Foundation models can take many forms; for example, $\widehat{m}_D$ could be some pre-trained model that is fine-tuned on the dataset $D$, or it can be an LLM that is supplied $D$ in-context.

**World model:** A postulated *world model* is summarized by a state space $\Phi$ and a mapping $\phi \colon \mathcal{X} \to \Phi$ that associates each input with some state $\phi(x) \in \Phi$. A dataset $D$ is *consistent* with the world model if for each $(x, y) \in D$, the output is a deterministic function of the state, $y = g(\phi(x))$ for some $g : \Phi \to \mathcal{Y}$.

### 2.1. Comparing foundation models to world models

There is a challenge in defining what it means for a foundation model to recover a world model: foundation models and world models operate in different spaces. A foundation model outputs a new predictive model from data, whereas a world model describes state structure implicit in data.

One approach would be to mechanistically probe the foundation model, e.g. by comparing its weight-level representations to the postulated states in the world model. However,

understanding the internal mechanisms of large models is challenging (Olah, 2022) and even then may not reflect how a model actually behaves on new data (Casper et al., 2023). Another approach is to study the model's behavior statically, on a single task (Toshniwal et al., 2022; Vafa et al., 2024), but this doesn't capture how foundation models are used in the real world: as tools for new tasks.

We take a different approach, motivated by the no-free-lunch theorem (Wolpert, 1996). Loosely speaking, the no-free-lunch theorem states that no learning algorithm can perform better than another one on average if any function could have generated the data it is applied to. Given limited data, learning algorithms must extrapolate to unseen inputs, and if any underlying function is possible, any such extrapolation must be equally good or bad. This means that every learning algorithm is better for *some* collection of possible functions — those functions that it tends to learn when extrapolating from limited data. The functions that a learning algorithm tends to learn represent its *inductive bias*.

The idea of inductive bias offers a connection between foundation models and world models. A world model is a restriction on the possible functions from inputs to outputs: only those that obey its state structure are possible. Consequently, a foundation model that has learned a postulated world model should have an inductive bias towards functions that obey the world model's state structure. For example, physicists may train a foundation model on sequences of planetary orbits. Since planetary orbits obey Newtonian mechanics, they might hope the model has an inductive bias toward functions of Newtonian mechanics (e.g. predicting the force vector between two planets).

We develop an *inductive bias probe* for testing whether a foundation model's inductive bias matches the postulated world model's state structure. The inductive bias probe repeatedly applies a foundation model to synthetic datasets consistent with the world model and studies the extrapolated functions together (Figure 9). In each such simulation, we do not calculate the "accuracy" of the resulting extrapolations since there is no one accurate function; multiple ways to extrapolate may be allowed by the true world model. Instead, we evaluate whether the extrapolations resemble those that are allowed by the true world model.

## 2.2. Special case: finite state space and binary outputs.

To provide more intuition for the inductive bias probe, we first consider the special case of a binary output $\mathcal{Y} = \{0, 1\}$ and a postulated world model with a finite state space $\Phi$. The two metrics we introduce in this setting are special cases of the general inductive bias probe defined in the next section.

The inductive bias probe evaluates whether a foundation model's inductive bias is towards a postulated world model.

At a high level, the probe repeatedly applies the foundation model to synthetic datasets consistent with the postulated world model and each time evaluates its predictions on held-out inputs. If the foundation model's inductive bias is towards the postulated world model, its extrapolations should have two properties. First, the foundation model's predictions should *respect state*: if two inputs $x, x'$ map to the same state ($\phi(x) = \phi(x')$), the foundation model should have the same predicted outputs ($\widehat{m}_D(x) = \widehat{m}_D(x')$) when applied across synthetic datasets. If not, it means that the foundation model fits functions that do not belong to the world model. Second, the foundation model's predictions should *distinguish state*: if two inputs $x, x'$ map to different states ($\phi(x) \neq \phi(x')$), the foundational model should typically have different predicted outputs ($\widehat{m}_D(x) \neq \widehat{m}_D(x')$) across synthetic datasets. If not, then the foundation model does not fit functions that fully cover the world model's allowable functions.

These properties can be measured using two metrics. Let $1(y, y')$ indicate whether $y = y'$. We specify a sampling distribution over consistent datasets $D \sim P_D$ and a sampling distribution over inputs $(X_i, X_j) \sim P_X \times P_X$. The foundation model's *inductive bias towards respecting state* (R-IB) is

$$\mathbb{E}_{X_i, X_j, D}[1(\widehat{m}_D(X_i), \widehat{m}_D(X_j)) \mid \phi(X_i) = \phi(X_j)]. \quad (1)$$

R-IB measures the similarity between the model's extrapolations on inputs in the same state under the postulated world model: higher R-IB indicates more similar predictions for the same states. The foundation model's *inductive bias towards distinguishing state* (D-IB) is

$$1 - \mathbb{E}_{X_i, X_j, D}[1(\widehat{m}_D(X_i), \widehat{m}_D(X_j)) \mid \phi(X_i) \neq \phi(X_j)]. \quad (2)$$

D-IB measures whether inputs that belong to different states under the postulated world model nonetheless receive consistently similar predictions by the foundation model: higher D-IB indicates more dissimilar predictions for different states. Figure 10 illustrates both metrics.

Together, R-IB and D-IB provide contrasting perspectives on a foundation model's implicit world model, analogous to precision and recall in binary classification. For example, while it is trivial for a foundation model to achieve high R-IB by making the same prediction on every input, its D-IB will suffer. Both metrics are needed to contrast a foundation model's inductive bias with the postulated world model.

In this sense, the inductive bias probe captures behavior of a foundation model that is not captured by standard probe tests (Nanda et al., 2023b), which measures how well a simple predictive model (e.g., a linear model) can predict state from a foundation model's intermediate representation. By contrast, the inductive bias probe directly analyzes how the foundation model behaves when adapted to synthetic tasks

from the postulated world model. When there are many distinct state mappings that are predictable from a foundation model's internal representation, standard probes cannot distinguish which is actually being used by the model. Moreover, the standard probe is sensitive to how state is mechanistically represented by the chosen world model. For example, Nanda et al. (2023b) find that different representations of the Othello game board (one based on the standard board and another that inverts the board based on whose turn it is) lead to different results by standard probes. By contrast, because inductive bias probes only depend on state equality, they are insensitive to equivalent representations.

To implement the inductive bias probe, a practitioner must supply a sampling distribution over consistent datasets $P_D$ and a sampling distribution over inputs $P_X$. In our experiments with a finite state space and binary outputs (see Section 4), we sample consistent datasets by assigning each unique state the output 0 or 1 uniformly at random.

### 2.3. Inductive bias probe

We now describe the inductive bias probe allowing for general outputs, state spaces, and tasks. For example, for sequences of two planets orbiting one another, the states could correspond to their relative positions, relative velocities, and the masses of each planet under Newtonian mechanics. We further introduce a collection of *admissible functions* on state $\mathcal{G}$ that govern the relationship between the state space and the output under the world model with each $g \in \mathcal{G} \colon \Phi \to \mathcal{Y}$. For example, in some settings, we may expect the output to vary smoothly with the state, in which case $\mathcal{G}$ could be the collection of $K$-Lipschitz functions. A dataset is now consistent with the world model if for each $(x, y) \in D$, $y = g(\phi(x))$ for some $g \in \mathcal{G}$.

Given a sampling distribution over consistent datasets $P_D$ and a sampling distribution over inputs $P_X$, the inductive bias probe repeatedly applies the foundation model to sampled datasets, and then evaluates its predictions on held-out inputs. It measures how *predictable* the foundation model's predicted outputs for one input are from those of another input across many synthetic datasets. The intuition is unchanged: inputs in "similar" states should be more predictable from one another than inputs from "different" states. We next formalize this property.

**Extrapolative predictability.** We further specify a family of predictors $\mathcal{H}$ with $h \in \mathcal{H}$ such that $h : \mathcal{Y} \to \mathcal{Y}$ and a loss function over outputs $\ell : \mathcal{Y} \times \mathcal{Y} \to \mathbb{R}_+$. We define the *extrapolative predictability* between two inputs as

$$\widehat{I}(x_i, x_j) = -\min_{h \in \mathcal{H}} \mathbb{E}_{D \sim P}[\ell(h(\widehat{m}_D(x_i)), \widehat{m}_D(x_j))], \quad (3)$$

which measures how predictable the foundation model's predicted outputs for one input are from the other. Higher

values of extrapolative predictability indicate higher levels of predictability. If a foundation model behaves as if it extrapolates based on the postulated world model, the extrapolative predictability should be larger for inputs with more similar states.

**Oracle foundation model.** As a calibration, we calculate the extrapolative predictability for an "oracle" foundation model that is given access to the true state space $\Phi$ and admissible functions $\mathcal{G}$. When applied to consistent dataset $D$, the oracle foundation model returns

$$m_D^* = \arg\min_{g \in \mathcal{G}} \frac{1}{|D|} \sum_{(x_i, y_i) \in D} \ell(g(\phi(x_i)), y_i). \quad (4)$$

(The loss function used here need not be the same as the loss function used to calculate extrapolative predictability.) The oracle extrapolative predictability is

$$I^*(x_i, x_j) = -\min_{h \in \mathcal{H}} \mathbb{E}_{D \sim P}[\ell(h(m_D^*(x_i)), m_D^*(x_j))]. \quad (5)$$

**Inductive bias towards the world model.** The inductive bias probe compares the foundation model's extrapolative predictability to that of the oracle. Specifically, the foundation model's *inductive bias towards the world model* is defined as, for any $0 \le \underline{s} \le \overline{s}$,

$$\mathrm{IB}(\underline{s}, \overline{s}) = \mathbb{E}_{X_i, X_j}[\widehat{I}(X_i, X_j) \mid \underline{s} \le I^*(X_i, X_j) \le \overline{s}]. \quad (6)$$

We calculate this quantity over a grid of values $0 = s_0 < s_1 < \cdots < s_m$, visualizing how $\mathrm{IB}(\underline{s}, \overline{s})$ varies over the grid. The foundation model's inductive bias towards the world model can be interpreted like a calibration curve: if the foundation model behaves like the oracle when applied to many small datasets, then $\mathrm{IB}(\underline{s}, \overline{s})$ should lie on the 45-degree line in this visualization (as illustrated in Figure 2).

R-IB and D-IB are special cases of Equation 6. Consider the case in which the output is binary, $\Phi$ is finite, and $\mathcal{G}$ is the collection of all mappings. Provided $P_D$ places positive probability on all possible consistent datasets and $\mathcal{H}$ is limited to the identity function, there are only two possible values for the oracle's extrapolative predictability, which occur when $\phi(x_i) = \phi(x_j)$ and when $\phi(x_i) \ne \phi(x_j)$. Consequently, the foundation model's inductive bias towards the world model reduces to R-IB in the former case (Equation 1) and D-IB (up to a sign change) in the latter case (Equation 2).

## 3. Orbital Mechanics

We illustrate these ideas by testing whether a transformer trained to predict the locations of planets in motion has recovered Newtonian mechanics.[1] We first train a model to predict the location of planets across solar systems.

---

[1]Our code is available at https://github.com/keyonvafa/inductive-bias-probes.

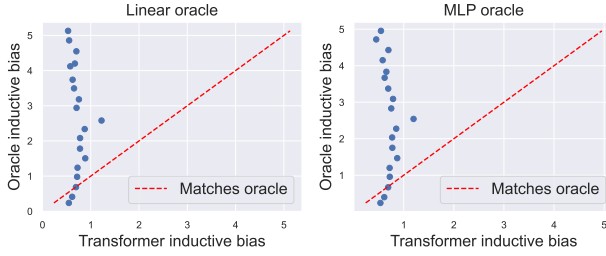

*Figure 2.* Inductive bias probe performance (Equation 6) for a transformer pretrained on orbital trajectories. A 45-degree line would indicate perfect inductive bias toward an oracle that extrapolates based on the Newtonian state vector.

Despite the model's ability to accurately predict the future trajectories of planets, the inductive bias probe reveals that it has a low inductive bias toward Newtonian mechanics. This is corroborated by the fact that when the model is fine-tuned to predict a planet's force vector — a cornerstone of Newtonian mechanics — its predictions imply a nonsensical law of gravitation. We find that the model has recovered piecemeal heuristics rather than a compact world model; it recovers different laws of gravitation on different data slices it is applied to.

**Background.** For centuries, astronomers and physicists have worked on predicting the orbits of planets around the sun. A groundbreaking model was offered by the astronomer Johannes Kepler in the 17th century. His model was based on geometric patterns: for example, that the orbit of each planet followed an ellipse with the sun at one of its foci. While the model could predict orbits with a near-perfect level of precision, it couldn't explain why the planets obeyed these geometric orbits or be applied to new problems beyond predicting trajectories.

Later, Isaac Newton expanded on this model using new laws of motion, now known as Newtonian mechanics. These laws involved computing properties of the set of planets in motion, such as their relative velocities and masses. Using these properties, he could derive Kepler's earlier laws for orbital trajectories, but also go beyond, understanding and formalizing other concepts like force and gravity.

From Kepler to Newton, scientists were able to move beyond good predictive models of sequences to a deeper understanding of them. In this section, we test whether a transformer that can predict sequences of orbital trajectories is merely a good sequence model, or whether it has also made the transition to providing a world model.

**Data and pre-training.** We first simulate a dataset of sequences, where each sequence describes planets in motion around a sun. To do this, we randomly sample initial conditions (e.g. the masses and positions of the planets and their

initial relative velocities) to target the shape of orbits observed in known exoplanets (Kipping, 2013). We simulate each planet's trajectory around the sun using Newton's laws of motion; because planet masses are much smaller than the sun's, planet interactions are minimal, so we omit them.

To convert orbits into sequences, we record $(x, y)$ coordinates of each planet and the sun across regular intervals, and interleave all the positions into a single sequence with 1,000 observations. This means that each sequence denotes a different solar system. We randomly sample half of the sequences to use 6-month time intervals between observations and use 1-week time intervals for the other half, using a special token at the beginning of the sequence to indicate the interval length. For example, in a solar system with $K$ planets, the first timestep encodes the interval length, the next $K$ observations are the $(x, y)$ coordinates for each planet at the first point in time, and the next $K$ are the coordinates for each planet the appropriate timestep later, etc. (We also considered using fixed-length intervals and found similar results.) We use a training set of 10M sequences and 20B tokens.

We train a 109M parameter transformer (Vaswani et al., 2017) to predict the next token of each sequence in the training set. We experimented between using a) continuous coordinates (and MSE loss) and b) discretized coordinates (with cross-entropy loss), finding the latter worked better. We discretize each position vector of each body in the solar system by creating 7K bins per coordinate $(x, y)$, where the coordinates spans from -50 to 50 AU. We train for 25 epochs using 8 H100 GPUs. See Appendix A for more training details.

We evaluate model predictions on held-out data. The model makes good predictions: its $R^2$ is above 0.9999, and it significantly outperforms baseline models that always predict the most recent position or the per-orbit mean (Table 8). It can also generate long orbits with a high degree of accuracy.

**Has the model recovered Newtonian mechanics?** The transformer's predictions reflect a very good sequence model. But has it recovered Newtonian mechanics? To test this, we note that Newtonian mechanics dictate that each observation in a sequence of orbits is governed by a state vector consisting of the masses, relative velocities, and relative positions of each planet. Given the current state of a trajectory, the next position of an orbit is deterministic. This is our world model; if a foundation model's inductive bias depends on Newtonian mechanics, it must be extrapolating based on this state vector.

We use the inductive bias probe described in Section 2 to assess the model's inductive biases. We create 100 synthetic datasets where the outputs are linear functions of the state of the sequence. We then fine-tune the transformer by training it to predict these functions. We measure

| Ground-truth law | $F \propto \frac{m_1 m_2}{r^2}$ |
|---|---|
| | Galaxy 1    $F \propto \left( \sin\left(\frac{1}{\sin(r-0.24)}\right) + 1.45 \right) * \frac{1}{\frac{1}{r} + m_2}$ |
| | Galaxy 2    $F \propto \cos(\cos(2.19 * m_1))$ |
| Estimated laws | Galaxy 3    $F \propto \cos(\sin(\frac{0.48}{m_1}))$ |
| | Galaxy 4    $F \propto \sin(r + 8569.2 + \frac{1}{m_1})$ |
| | Galaxy 5    $F \propto \cos(\cos(e^{m_2}))$ |

*Table 1.* Force equations recovered via symbolic regression of a transformer pretrained on orbital data and fine-tuned to different galaxy samples. The model recovers different equations for each sample, never recovering the true law.

the model's extrapolative predictability across inputs (Equation 3) by considering $\mathcal{H}$ to consist of the identity and the loss function $\ell$ to be MSE. We evaluate Equation 6 by comparing the model to an oracle that extrapolates based on state directly (we consider both linear models and 2-layer neural networks for the oracle, finding similar results). The inductive bias toward simple functions of Newtonian state is poor; see Figure 2 for a visualization. In other words, the model's inductive bias is not toward Newtonian state; when it has to extrapolate, it makes similar predictions for orbits with very different states and different predictions for orbits with very similar states. For implementation details and ablations, see Appendix B.1.

To understand the degree to which the model fails to apply Newtonian mechanics, we test its ability to predict specific quantities derived from Newtonian mechanics. Specifically, we consider each planet's force vector, a simple transformation of state given by Newton's law of gravitation: $\mathbf{F} = G \frac{m_1 m_2}{||\mathbf{r}||^2} \mathbf{e}_r$, which relates the force $\mathbf{F}$ between a planet and the sun to their masses $m_1, m_2$ and their squared distance $||\mathbf{r}||^2$ (in the direction $\mathbf{e}_r$ of its relative position). The force vector can be computed for each observation in a sequence; force is a simple transformation of state, so the predictions of a model that has recovered Newtonian mechanics should obey this law.

We test this by creating a sequence-to-sequence dataset where each input is a trajectory and each output is the force vector $\mathbf{F}$ on the planet implied by the state of the orbit. We first fine-tune the pretrained transformer to predict the force vector on orbits from our solar system, providing 1% of the true forces as training data. Figure 1 shows these force predictions are poor. To assess how close the model is to recovering Newton's law of gravitation, we further fine-tune it to predict the force magnitude on a larger dataset of 10K solar systems. We then perform a symbolic regression (using the *PySR* software (Cranmer, 2023)) of the predicted force magnitudes on the true values of $m_1, m_2$, and $r$. A symbolic regression is a method to search for a symbolic expression that optimizes a regression-like objective (Cranmer

et al., 2020). When the symbolic regression is applied to the transformer's predictions, the physical law is nonsensical (Figure 1). In contrast, an oracle trained on the true state predicts the force vectors well and a symbolic regression recovers the true physical law (Figure 6 in Appendix C). See Appendix C for implementation details and Appendix D for a similar experiment with LLMs.

How can a model perform so well at predicting orbit locations without having inductive biases towards the laws of physics that govern them? We study this question by applying the fine-tuned model's force predictions to five different sets of randomly sampled galaxies (each consisting of many solar systems). We then perform a symbolic regression on the force magnitude for each sample. The symbolic regression finds a different implied law of gravitation for each sample (Table 1). In contrast, the oracle trained on true state recovers the same (correct) law for each galaxy. These results show that rather than building a single universal law, the transformer extrapolates as if it constructs different laws for each sample.

## 4. Other Applications

We now apply the inductive bias probe to evaluate the extent to which foundation models obey known world models in other domains. Evaluating world models requires studying domains where there's a state structure and ground-truth state is known. We study two such types of datasets: lattice problems and the board game Othello.

**Lattice.** One common type of structure to assess models against is spatial structure, or lattices (Vafa et al., 2024; Liu et al., 2022). We study a lattice setting that simulates an agent moving along a line segment with a finite number of positions. There is a true state space consisting of $S$ states: $\Phi = \{1, 2, \ldots, S\}$. The language $x$ consists of sequences with three tokens: $\Sigma = \{L, \perp, R\}$. The initial state of the sequence is 1. For a token $\sigma = R$, the state increases by 1, while the state decreases by 1 for $\sigma = L$ and stays the same for $\sigma = \perp$. When the state is 1, the state is at the boundary, so $\sigma = L$ is not a valid token; similarly, when the state is $S, \sigma = R$ is not a valid token. All tokens are valid for all other states. We randomly generate sequences of length 100 over the language by sampling a move uniformly at random over the set of valid moves for each timestep. We consider different versions of the lattice problem, varying the number of states from 2 to 5. We consider sequences taken from a training set containing 10M tokens, along with 100k hold-out tokens.

**Othello.** We also study the board game Othello, a common testbed for evaluating the world models of sequence models (Li et al., 2023; Nanda et al., 2023b; Hazineh et al., 2023; Vafa et al., 2024). The game consists of two players taking

| | Pre-training | Lattice (5 States) | | Othello | |
|---|---|---|---|---|---|
| | | R-IB (↑) | D-IB (↑) | R-IB (↑) | D-IB (↑) |
| **RNN** | Untrained | 0.346 (0.026) | 0.749 (0.027) | 0.228 (0.016) | 0.990 (0.002) |
| (Elman, 1990) | NTP trained | 0.574 (0.026) | 0.803 (0.032) | 0.632 (0.023) | 0.797 (0.023) |
| **LSTM** | Untrained | 0.456 (0.028) | 0.718 (0.031) | 0.438 (0.030) | 0.681 (0.031) |
| (Hochreiter, 1997) | NTP trained | 0.782 (0.021) | 0.921 (0.030) | 0.563 (0.030) | 0.610 (0.034) |
| **Transformer** | Untrained | 0.268 (0.027) | 0.742 (0.028) | 0.708 (0.022) | 0.843 (0.021) |
| (Vaswani et al., 2017) | NTP trained | 0.483 (0.031) | 0.677 (0.034) | 0.703 (0.025) | 0.624 (0.033) |
| **Mamba** | Untrained | 0.260 (0.026) | 0.771 (0.027) | 0.303 (0.016) | 0.929 (0.009) |
| (Gu & Dao, 2023) | NTP trained | 0.571 (0.023) | 0.866 (0.029) | 0.682 (0.021) | 0.728 (0.027) |
| **Mamba-2** | Untrained | 0.244 (0.026) | 0.785 (0.026) | 0.468 (0.019) | 0.896 (0.016) |
| (Dao & Gu, 2024) | NTP trained | 0.617 (0.021) | 0.864 (0.029) | 0.653 (0.022) | 0.694 (0.029) |

*Table 2.* The *inductive bias towards respecting state* (R-IB) and *inductive bias towards distinguishing state* (D-IB) metrics (1 is perfect performance, 0 is equivalent to noninformative model). "NTP-trained" represents a model pre-trained on next-token prediction, while "untrained" refers to a model trained on the same synthetic tasks, initialized from scratch.

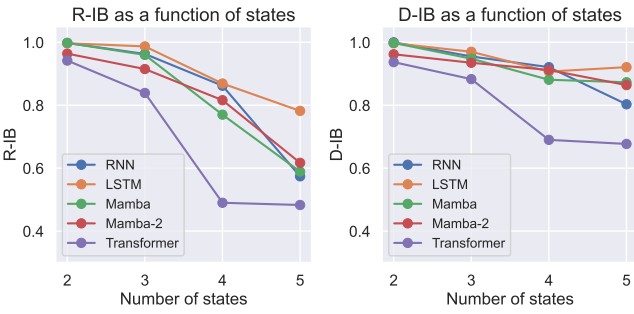

*Figure 3.* Inductive bias probe results (R-IB and D-IB) for the lattice problem as a function of the underlying number of states. A different model is pre-trained on data consistent with each number of states and its inductive bias for that state structure is recorded using the metrics in Section 2.

turns placing tiles on an 8x8 board. Each game of Othello is tokenized into a sequence of at most 60 moves, where each token indicates which of the 60 squares the most recent tile was placed on (the middle four tiles are always occupied). The true state space $\Phi$ corresponds to all 8x8 boards and the mapping $\phi$ converts game sequences into states. We consider game sequences taken from a training set containing 20M games, along with 3.8M hold-out games.

**Models.** We study the properties for five classes of pre-trained sequence models: RNNs (Elman, 1990), LSTMs (Hochreiter, 1997), transformers (Vaswani et al., 2017), Mamba (Gu & Dao, 2023), and Mamba-2 (Dao & Gu, 2024). We train each model using next-token prediction for each domain. By way of comparison, we also compare these pretrained models to untrained models that fine-tune from a random initialization. See Appendix A for details.

All pre-trained models perform well at next-token prediction, generating outputs that appear to obey state. Following

Toshniwal et al. (2022), we measure the fraction of a model's top predictions that are legal in the underlying state. Table 7 in Appendix G shows the results. All models do very well across all datasets, e.g. every model's top prediction is legal ≈ 90% of the time for Othello and legal 100% of the time for a lattice problem with five states.

**Inductive bias probe results.** We measure each model's inductive bias using the procedure from Section 2 to assess the inductive bias of these models. The procedure involves fine-tuning each model to small datasets of randomly generated outputs and assessing whether the model's inductive bias — as measured by its extrapolations — obeys state structure. We use the discrete version of the procedure for both models.

The results for the lattice problem are depicted in Figure 3. While models have high inductive biases when the number of states is small, as the number of states increases, the inductive biases drop off. Notably, the transformer model consistently does worse than the other models, all of which have architectures based on recurrent or state-space models. The results for Othello are depicted in Table 2. Here, all models perform worse than on the lattice problems, indicating poor inductive bias. Despite generating legal moves nearly 100% of the time when pretrained to play Othello, these models don't use the board as an inductive bias on new tasks.

To understand the implications of these results, we study how different models transfer to new functions of state (the board). Specifically, we take the Othello dataset and construct new sequence-to-sequence datasets. The input sequence for each dataset is the original game transcript, and we consider three different output transformations that are functions of state. In "Majority Tiles", each element of the output is 1 or 0 indicating where there are more black or white tiles in the board implied by the sequence so far. In "Board Balance", each element of the output sequence

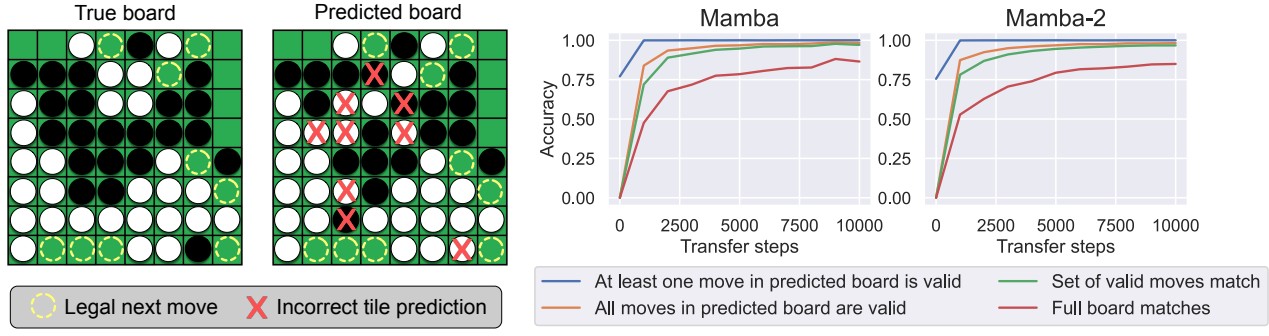

*Figure 4.* On the left, a true Othello board implied by a sequence, and on the right, the predicted board from a model fine-tuned to predict boards. Although the prediction has errors, the set of predicted next tokens exactly matches the true board. On the right, metrics about board reconstruction during fine-tuning. Consistently, even as Mamba models struggle to recover full boards, they recover them well enough such that the sets of valid next moves match those in the true boards.

indicates whether black has more pieces in the top half of the board or in the bottom half of the board. Finally, in "Edge Balance", the output measures whether black has more pieces along the edge squares of the board. Each of these functions is a deterministic function of state (the board), so foundation models that have inductive bias toward state should be better at transfer. The results are depicted in Table 6. The last row shows the (unsigned) correlation for each metric and the ratio, $\frac{\text{R-IB}}{1-\text{D-IB}}$ that summarizes the inductive bias measures in Table 2. There is strong correlation across all metrics; models that do better on inductive bias metrics transfer better to these functions of state.

**What are the inductive biases?** These results show that models can perform well at predicting token sequences without appearing to learn the underlying world model. This raises the question: If a foundation model's inductive bias isn't toward a given world model, what is it toward?

Here, we consider one hypothesis motivated by the next-token pretraining objective: that when foundation models are applied to new tasks, they group together sequences with distinct states for which the set of legal next tokens are nevertheless equivalent. For example, in the board game Othello, two distinct boards can have the same set of allowable next moves. Therefore, a model's inductive bias might be toward boards with the same sets of allowable next moves rather than the true board itself.

To first demonstrate this concept with Othello, we fine-tune a foundation model originally pretrained to perform next-token prediction on 1M games to now predict the true board of each sequence. We record two metrics when we fine-tune: 1) whether the predicted board exactly matches the true board, and 2) whether the set of valid moves in the predicted board matches the set of valid moves in the true board. The results are depicted in Figure 4: surprisingly, even when the predicted board is incorrect, the set of legal moves

frequently matches the set of legal moves from the true board. Rather than recovering the full board, the foundation model is often recovering "enough of" the board to calculate legal next moves.

To quantify this hypothesis generally, we modify the inductive bias probe to test whether a model's inductive bias is toward *next-token partitions* of state. Recall that D-IB measures how similar extrapolations for two points with different states are one from another. If a model is extrapolating based on which next-tokens are legal, sequences in different states that happen to have the same legal next tokens will have more similar predictions than sequences in different states that have different legal next tokens.

Specifically, let $q$ denote the *next-token coarsening* of the state space such that $q(x) = q(x')$ if and only if $\text{NextTokens}(\phi(x)) = \text{NextTokens}(\phi(x'))$, where $\text{NextTokens}(s)$ is the set of valid next tokens for state $s$. We decompose D-IB into two quantities. First, define $\text{Same}(X_i, X_j)$ as the event that $\phi(X_i) \neq \phi(X_j)$ but $q(X_i) = q(X_j)$. We then define,

$$\text{D-IB}_{q=} = 1 - \mathbb{E}\left[1(\widehat{m}_D(X_i), \widehat{m}_D(X_j)) \mid \text{Same}(X_i, X_j)\right],$$

which measures how predictable the extrapolations for inputs associated with different states that have the *same* legal next tokens are. Similarly, define $\text{Diff}(X_i, X_j)$ as the event that $\phi(X_i) \neq \phi(X_j)$ and $q(X_i) \neq q(X_j)$. Analogously,

$$\text{D-IB}_{q\neq} = 1 - \mathbb{E}\left[1(\widehat{m}_D(X_i), \widehat{m}_D(X_j)) \mid \text{Diff}(X_i, X_j)\right],$$

which measures how predictable the extrapolations for inputs associated with different states that have *different* legal next tokens are. If distinct-state inputs with the same legal next tokens are more predictable than distinct-state inputs with different legal next tokens (i.e., $\text{D-IB}_{q=} < \text{D-IB}_{q\neq}$), then it suggests the model extrapolates based on the next-token partition rather than the true board state.

We compute these refined metrics for lattice and Othello. Each has a natural definition of legal next moves (corresponding to boundaries and game rules). The results are depicted in Table 9. For all models, the gap between D-IB$_{q=}$ and D-IB$_{q\neq}$ is statistically significant, suggesting that models are grouping together distinct states with the same sets of legal next tokens.

## 5. Related Work

This paper studies whether predictive models form world models (LeCun, 2022). One strand of world model research studies whether the outputs of a fixed model accord with a known world model by studying the fixed model's outputs (Vafa et al., 2024). For example, one way that Toshniwal et al. (2022) and Li et al. (2023) study world models is by assessing whether a model trained on sequential game data always predicts legal moves in the underlying game. The question we study is a different yet related question: rather than studying the world model properties of a fixed model, we study what it means to test if a *learning algorithm* — a foundation model — has a world model embodied in it.

Another strand of the literature assesses whether a model's parametric *representations* encode world models (Abdou et al., 2021; Patel & Pavlick, 2022; Gurnee & Tegmark, 2023; Nanda et al., 2023a). For example, a common method uses probes or sparse autoencoders (SAEs) (Cunningham et al., 2023; Trenton Bricken et al., 2023) to assess whether an intermediate representation used by a neural network is predictive of state (Hewitt & Liang, 2019; Li et al., 2021; Abdou et al., 2021; Jin & Rinard, 2023; Li et al., 2023; Spies et al., 2024; Karvonen, 2024). However, there are open questions about the reliability of probes (Belinkov, 2022), such as appropriate function complexity (Alain & Bengio, 2018; Cao et al., 2021; Li et al., 2023). Our method sidesteps these issues by asking how a model *learns*, rather than what's encoded in its fixed representations. Closely related to us, jylin04 et al. (2024) and Nikankin et al. (2024) find that a GPT model trained on Othello and math tasks, respectively, performs internal computations corresponding to "bags of heuristics" rather than a coherent world model. While our procedures differ in aim, these findings support our analysis of the Othello model relying on heuristics, rather than state, as its inductive bias (McCoy et al., 2019).

Complementary to the internal focus of mechanistic interpretability, other research uses behavioral probes to study a model's ability to synthesize knowledge, a methodology closer to our own. For instance, recent work demonstrates that LLMs can infer and internalize latent knowledge from disparate information seen during training, and then apply this inferred knowledge to downstream tasks (Berglund et al., 2023; Treutlein et al., 2024). Our inductive bias probe provides a framework for testing whether such emergent knowledge constitutes a robust world model.

The methodology in this paper is rooted in the observation that generative models can reach the same generated outputs in different ways. This is related to the Rashomon effect (also referred to as model multiplicity) for predictive models, where there can exist many different models that achieve similar performance on a predictive task (Breiman, 2001; Marx et al., 2020; D'Amour et al., 2022; Black et al., 2022). The literature on causal representation learning suggests that models should learn representations corresponding to the true causal mechanisms to generalize robustly to new tasks (Schölkopf et al., 2021; Bengio et al., 2019). Our method is based on a similar motivation, as it studies the properties of a foundation model not by its generations for one task but rather its inductive bias. A model that has learned a true world model should find new tasks "informationally close" (Achille et al., 2021) and adapt easily.

Recent work developing foundation models in scientific domains such as protein folding, gene regulation, and molecular chemistry (Chowdhury et al., 2022; Benegas et al., 2023; Boiko et al., 2023; Jablonka et al., 2024) use predictive models as stepping stones toward uncovering deeper principles. Our orbital mechanics example relates specifically to the large body of work studying AI and physics (Hao et al., 2022; Wu & Tegmark, 2019). It is most closely related to works studying whether AI models can uncover physical laws (Chen et al., 2022; Belyshev et al., 2024; Kansky et al., 2017; Udrescu & Tegmark, 2020; Iten et al., 2020). Most closely to us, Lemos et al. (2023) demonstrate that Newton's gravitational law can indeed be recovered from a graph neural network trained on orbital data by modifying the model architecture to explicitly impose Newton's laws of motion as inductive biases. We find that a transformer without Newton's inductive biases does not recover the gravitational law, but imposing domain-specific inductive biases is a promising approach to improving these models. We adopt general tools from this literature — such as using symbolic regressions for interpretability — to study the inductive biases of algorithms (Liu & Tegmark, 2021; Wu & Tegmark, 2019).

## 6. Conclusion

The promise of foundation models is that sequence prediction can uncover deeper understanding of underlying mechanisms. We develop a framework for evaluating whether a foundation model has learned a postulated world model by measuring its inductive biases when transferring to new tasks. Our empirical results reveal that while many sequence models excel at next-token prediction tasks, they often have limited inductive bias toward genuine world models. Rather than learning coherent world models, we find that these models may be relying on coarsened state representations or non-parsimonious representations.

## Acknowledgments

Keyon Vafa is supported by the Harvard Data Science Initiative. Peter Chang is supported by the NSF CSGrad4US Fellowship.

## Impact Statement

This paper presents work whose goal is to advance the field of Machine Learning. There are many potential societal consequences of our work, none which we feel must be specifically highlighted here.

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

## A. Model and Training Details

We use the following specifications for each model:

- RNN (Elman, 1990): For Othello, We use 6 uni-directional RNN layers with 768 embedding dimensions. For the lattice experiments, the architecture is the same except we use only 2 layers because it optimizes to better in-sample and out-of-sample loss.

- LSTM (Hochreiter, 1997): We use the same specification as for the RNN, except we use LSTM layers.

- Transformer (Vaswani et al., 2017): We use a transformer decoder architecture, with 12 layers, 12 attention heads, and 768 embedding dimensions.

- Mamba (Gu & Dao, 2023): We first encode inputs with a 768-dimension embedding layer. We then pass inputs through 24 Mamba layers (analogous to 12 layers in a transformer due to how Mamba layers are defined). We use 768 embedding dimensions, 16 for the SSM state expansion factor, 2 for the block expansion factor, and 4 for the convolutional width.

- Mamba-2 (Dao & Gu, 2024): We use the same architecture as for Mamba except the mixer in each block is a Mamba-2 module. We use the same specifications as well: 768 embedding dimensions, an SSM state expansion factor of 16, a block expansion factor of 2, and a convolutional width of 4.

We use Adam (Kingma & Ba, 2014) to optimize each model. We use a learning rate of 6e-4 and decay the learning rate with 2000 warmup iterations. We use weight decay of 0.1 and gradient clipping at 1 for each model. When we pre-train models on next-token prediction, we include a head to predict next tokens (tying its parameter weights to the initial embedding layer parameters).

For physics dataset generation, we use the following sampling strategy. For each solar system, we sample the number of planets from $\text{Unif}([1, 2, \ldots, 10])$, the eccentricity from a $\text{Beta}(\alpha = 0.867, \beta = 3.03)$ following Kipping (2013), the semi-major axis from $\text{Unif}(0.3, 42)$, in astronomical units (AU), the mass of each planet from $\text{LogUniform}(10^{-7}, 10^{-3})$, the mass of the star from $\text{Unif}(0.5, 5)$. These distributions ensure that our solar system is within the training distribution of the model. In order to generate sequences, we randomly generate initial conditions and solve Kepler's equation to obtain each trajectory.

## B. Metric Implementation Details

### B.1. Physics

To compute the empirical approximations of Equation 6, we follow the following procedure. First, we create 100 datasets of 100 examples, $D_1, \ldots, D_{100}$. For each dataset $D_i$, we sample 100 sequences uniformly at random among the set of data points and consider their corresponding sequences of state-vectors. First, we randomly sample 50 matrices of dimension $(6 \times 1)$ from standard Gaussian. We consider the linear projection of each state-vector using each of the 50 matrices, and choose the one that maximizes the Spearman correlation between pairwise Euclidian distances in the 6D state space and the projected 1D space. We randomly sample a projected point from each sequence, leading to $D_i$ of size 100. We then fine-tune a model separately for each dataset, resulting in 100 fine-tuned models $\hat{m}(\cdot; D_1), \ldots, \hat{m}(\cdot; D_{100})$. We then calculate the associated prediction functions across all inputs $x_i$ from the same hold-out dataset, resulting in new datasets of the form $\{(x_i, \hat{m}(x_i; D_1))\}, \ldots, \{(x_i, \hat{m}(x_i; D_{100}))\}$.

To compute the metrics, we first randomly sample 2,000 examples from all inputs, $x_{k_1}, \ldots x_{k_{100}}$, compute the pairwise Euclidean distance among the Oracle (a linear map or a 2 layer MLP with 5 nodes in each hidden layer) predictions on the inputs, and divide the range of predictions into 20 equally-spaced bins. For all the points that lie in each bin, we compute the mean pairwise Euclidean distance among the model predictions. The resulting figure is shown in Figure 2.

**Modified setup.** To further validate our findings and test the robustness of our inductive bias probe, we conducted additional experiments with modified training configurations and evaluation protocols.

The pretraining data for the orbital simulations in Section 3 is constructed to resemble our universe: each sampled galaxy consists of one sun and up to 10 planets, where the mass of the sun is much larger than that of the planets. In this setting, the

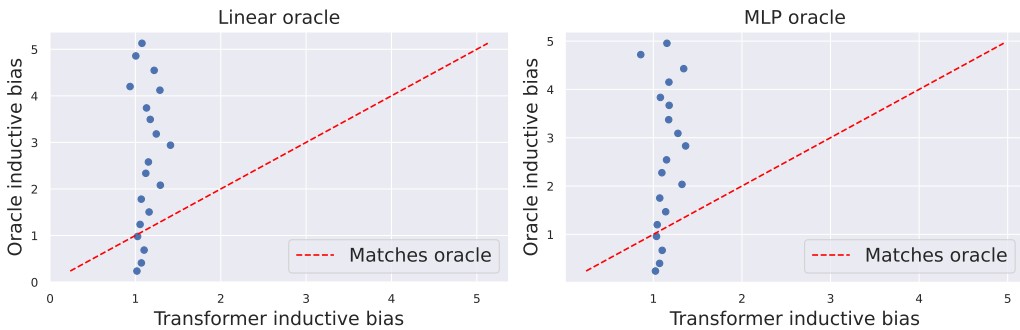

*Figure 5.* Modified inductive bias probe performance for a transformer pretrained on two-body systems. For these modified metrics, when the model extrapolates it doesn't extrapolate to brand new sequences but rather sequences that have been partially observed during training.

sun's movements are negligible, and interactions between planets are also minimal and ignored for computational reasons. Still, it's possible that a more restrictive setting — where there are only two masses in each solar system, each with similar masses — would result in better performance.

To test this, we pretrained a transformer on 10 million two-body systems, over 10 epochs, in the center-of-mass reference frame, with both masses sampled from a uniform distribution with range 1e-4 and 1e-2 solar masses, and other parameters unchanged from Section 3. We found nearly identical to the ones in the main text:.

We also considered a simpler version of the inductive bias metrics. Instead of using the extrapolations for out-of-sample sequences, we used the same trajectories for training and extrapolation. Specifically, we fine-tuned the model on two randomly observed output points per sequence, and then extrapolated the model on the rest of the trajectory. This would make it easier for the model to capture constant terms (e.g. masses) that do not change during the trajectory.

Figure 5 shows the results for the model trained on two-body data and evaluated using the modified inductive bias probe. The model still exhibits poor inductive bias toward Newtonian mechanics, with points clustered away from the 45-degree oracle line. Symbolic regression on force magnitude predictions yields the nonsensical equation $F \propto m_1 \times \frac{1}{\exp(r)}$, failing to recover Newton's law.

### B.2. Lattice and Othello

To compute the empirical approximations of Equation 1 and Equation 2, we follow the following procedure. First, we create 100 datasets of 100 examples, $D_1, \ldots, D_{100}$. For each dataset, we sample sequences uniformly at random among the set of data points and sample outputs from a Bernoulli(0.5) distribution. In our construction we make sure that any two sequences with the same state are mapped to the same output variable. We then fine-tune a model separately for each dataset, resulting in 100 fine-tuned models $\hat{m}(\cdot; D_1), \ldots, \hat{m}(\cdot; D_{100})$. We then calculate the associated prediction functions across all inputs $x_i$ from the same hold-out dataset, resulting in new datasets of the form $\{(x_i, \hat{m}(x_i; D_1))\}, \ldots, \{(x_i, \hat{m}(x_i; D_{100}))\}$.

To compute the metrics, we first randomly sample 2,000 examples from all inputs, $x_{k_1}, \ldots x_{k_{100}}$, then measure the average predictive loss for all pairs $(x_{k_i}, x_{k_j})$ with the same state, $\phi(x_{k_i}) = \phi(x_{k_j})$ (R-IB):

$$\text{R-IB} \approx \mathbb{E}_{D \sim D^{test}} \left[ \mathbb{E}_{i,j:\phi(x_{k_i})=\phi(x_{k_j})} \left[ m(x_i; D) = m(x_j; D) \right] \right] \tag{7}$$

and the average predictive loss for all pairs $(x_{k_i}, x_{k_j})$ with different states, $\phi(x_{k_i}) \neq \phi(x_{k_j})$ (D-IB):

$$\text{D-IB} \approx 1 - \mathbb{E}_{D \sim D^{test}} \left[ \mathbb{E}_{i,j:\phi(x_{k_i})\neq\phi(x_{k_j})} \left[ m(x_i; D) = m(x_j; D) \right] \right] \tag{8}$$

We rescale them so that the value of 0 corresponds to perfect accuracy and 1 corresponds to random guessing (large values for both indicate the model exhibits stronger inductive bias towards the state).

For the lattice example, we use a state space consisting of k states: $\Phi = \{1, 2, \ldots, k\}$. The inputs $x_i$ for extrapolation are taken from 1,000 random sequences of valid moves, each of length 100, for a total of 100,000 sub-sequences of moves. Our

procedure for Othello follows the same steps as for the lattice example, except the state is a 64-dimensional board instead of a single categorical variable.

Note that for Othello, if we randomly sample sequences from game transcripts, it is exceedingly likely that we end up with a dataset in which two sequences lead to the same state if and only if they are permutations of one another. This implies that a non-sophisticated model that detects unique permutations of sequences would appear to have high inductive bias towards the state. To prevent this, we first construct all valid Othello game openings of depth 10, randomly choose a board that appears many times in this dataset, then use all possible valid permutations of any sequence of moves that leads to that board as our input dataset. Note that since all sequences will be permutations of one another, the non-sophisticated model would no longer be able to distinguish different states. We end up with an input dataset of 210 Othello openings, each of length 10, for a total of 2,100 subsequence of moves.

## C. Force Prediction Implementation Details

Here we describe more implementation details for the force prediction experiments.

**Force vector prediction.** To create Figure 1, we fine-tuned the transformer to predict force vectors in two-body gravitational systems. We keep force vectors as continuous, and normalize the force vectors in each sequence so the maximum force vector in each sequence is unit length. We specifically fine-tune the model on the 8 sequences consisting of the trajectories in our solar system, randomly using 1% of the observations in each sequence as labeled force vector data for the model. We fine-tune the model to minimize MSE for 10,000 steps. We consider a learning rate grid between 1e-6 and 5e-4, finding that 2e-4 has the best validation loss. We keep the checkpoint with the lowest held-out loss. The model is then extrapolated to make predictions across the remainder of the points in each sequence.

For comparison, we perform the same procedure for an oracle model that predicts force vectors based on the true state matrices. Specifically, using the same sampling procedure, the oracle fits a $k$-nearest neighbor model with $k = 2$ based on Euclidean distance with the true state. We then use this model to predict force vectors for the remainder of the points in the solar system. The oracle predictions are depicted in Figure 6. These results show that it is feasible for a model to make accurate predictions if it is extrapolating based on the correct world model.

**Force magnitude prediction and symbolic regression.** We use a symbolic regression to assess how close the recovered force equation is to the true law. To simplify, we use the force magnitude rather than the full vector for these experiments (the vector is always in the direction of the sun). Here, we don't normalize the force magnitudes per solar system in order to preserve the force magnitude's dependence on the sun's mass.

We start by creating a training set that includes 9K two-body problems sampled using the sampling strategy in Appendix A. We create a test set of 1K sequences of two-body problems. We additionally ensure that the model is always extrapolating to sequences where it has seen partial information by adding two randomly sampled timestep observations of each test set sequence to the training set. Because $F \propto m_1 m_2/r^2$, this means that the only factor changing within the sequence is the $r^2$ term. Additionally, instead of imputing predictions on the full test set, we select the 5,000 timesteps across the 1,000 sequences that have the most similar states to states in the training set (using Euclidean distance). This ensures that the model is extrapolating to states that are similar to the ones it is trained on.

We fine-tune the transformer on the training set for 10,000 steps with a batch size of 64, keeping the checkpoint with the lowest held-out MSE. We impute the model's predictions on 1,000 randomly sampled points from the test set. We fit a symbolic regression to these predictions using the PySR library (Cranmer, 2023). Specifically we constrain our search space to have a max size of 20 and we consider two binary operators (addition and multiplication) along with 4 unary operators (sine, cosine, exponentiation, and inverse). We use a loss function that applies 0 penalty if the model is within 1e-8 of the magnitude and otherwise penalizes based on the absolute distance. We choose the model with the best score across three random restarts of 100 iterations each. We perform this symbolic regression procedure five times, each time randomly sampling 1,000 different points from the test set to correspond to a different galaxy. The symbolic regression returns different equations for each sample, never recovering the true law (Table 1).

To make sure this procedure is feasible when a model is extrapolating based on true state, we also consider an oracle model that is given true state. Specifically, we use the same data and fit a $k$-nearest neighbor model with $k = 2$ based on Euclidean distance to the true state. We then use this model to predict the same held-out points as above and fit symbolic regressions in the same manner. In contrast to the transformer results in Table 1, we find that this procedure recovers the true gravitational

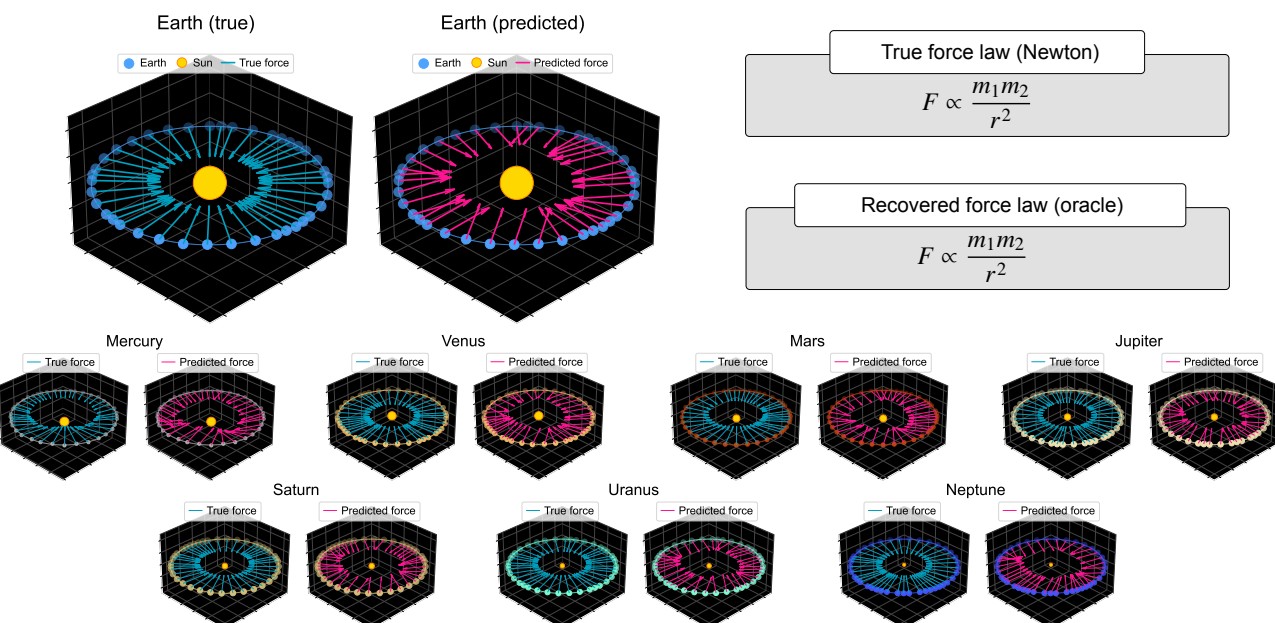

*Figure 6.* Each pair of panels illustrates the trajectory of a planet in the solar system and its gravitational force vectors, comparing the true Newtonian forces (left) to the predicted forces from an **oracle model** that predicts force vectors based on the true state matrices. A symbolic regression recovers the true gravitational law from its predictions.

| **Ground-truth law** | | $F \propto \dfrac{m_1 m_2}{r^2}$ |
| --- | --- | --- |
| **Estimated laws** | o3 | $F \propto m_1$ |
| | Claude Sonnet 4 | $F \propto \dfrac{1}{m_2 - 0.50}$ |
| | Gemini 2.5 Pro | $F \propto m_1$ |

*Table 3.* Force equations recovered via symbolic regression of LLMs predicting force magnitudes.

law for all five sampled galaxies.

## D. LLM Physics Experiments

Throughout this paper, we train foundation models on domain-specific data. Here, we consider large language models (LLMs) as foundation models for physics. While LLMs aren't trained on the same domain-specific trajectories we use, they are trained on large quantities of text that contain information about physics and orbital trajectories.

We consider three advanced reasoning models: o3 (from OpenAI), Claude 4 Sonnet (from Anthropic), and Gemini 2.5 Pro (from Google). Fine-tuning these models is infeasible because they're proprietary and running the full inductive bias probe is expensive because it involves applying the model to many new datasets. Instead, we run a small-scale experiment assessing each model's ability to predict the force magnitude of orbital trajectories. Rather than fine-tuning models, we provide them with information in-context, and study their extrapolation behavior. Specifically, we sample 5 random solar systems with 450 observations each. For each solar system, we provide each LLM with a prompt that describes the structure of the data, also including the true force magnitudes for 10 randomly selected observations. We instruct the LLM to predict the outputs for the remaining data points (we do not provide any information in the prompt indicating that the outputs correspond to forces). See Figure 8 for an example of the prompt.

We collect the magnitude inferences for each solar system (2,250 observations per LLM). Figure 7 shows the predicted force magnitudes for each solar system for each model. Most of the results are poor, which is further corroborated by symbolic

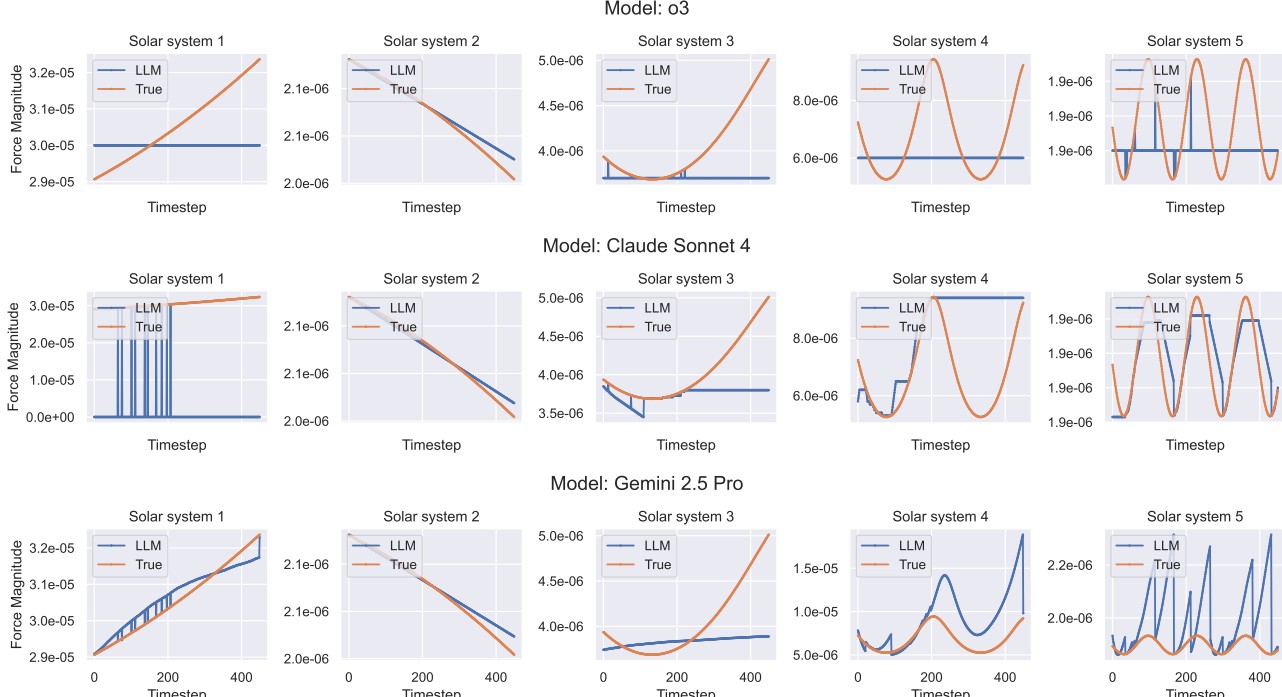

*Figure 7.* Comparing LLM magnitude predictions to the true magnitude across timesteps for 5 randomly sampled solar systems. Each LLM is provided the full trajectory and a random 2% sample of force magnitudes, and is prompted to impute the remaining outcomes.

regressions (Table 3). Interestingly, the symbolic regressions are simpler than the ones found for the domain-specific foundation models. However, this may be due to differences in experimental setup (e.g. using fewer solar systems for the LLM due to cost concerns).

---

**LLM Prompt**

```
You are a physics expert. You are given a sequence of coordinates and outcomes. The
coordinates are the positions of a planet in a 2-body solar system. The planet is
orbiting the sun. The sun is at the origin.

Here is a sequence of observations. Some of them are unknown. Your job is to predict
the outcomes for the unknown timesteps.

Timestep: 0, Coordinates: (-26.08, -6.98), Outcome: Unk
Timestep: 1, Coordinates: (-26.08, -6.99), Outcome: Unk
Timestep: 2, Coordinates: (-26.06, -7.01), Outcome: 2.907672751462087e-05
Timestep: 3, Coordinates: (-26.06, -7.04), Outcome: Unk
Timestep: 4, Coordinates: (-26.05, -7.05), Outcome: Unk
Timestep: 5, Coordinates: (-26.04, -7.08), Outcome: 2.90934076647451386e-05
Timestep: 6, Coordinates: (-26.04, -7.09), Outcome: Unk
Timestep: 7, Coordinates: (-26.02, -7.12), Outcome: Unk
Timestep: 8, Coordinates: (-26.02, -7.14), Outcome: Unk
Timestep: 9, Coordinates: (-26.01, -7.16), Outcome: Unk
...
Timestep: 449, Coordinates: (-20.28, -15.66), Outcome: Unk

You can reason all you'd like, but your answer should end with "ANSWER: " followed by
the predicted outcomes for all of the timesteps, even the unknown ones. You should
structure your predictions as a dict, where each key is a timestep and each value is
the prediction. You should make predictions for all of the timesteps, even the ones
that are known.

Here is an example of the output format:
ANSWER: {
    0: 1.0e-8,
    1: 1.0e-8,
    2: 1.0e-8,
    ...
    449: 1.0e-8,
}
```

*Figure 8.* Example prompt used in the LLM physics experiments.

## E. Inductive Bias Ablations

On the Othello dataset, we perform ablation of the IB metrics on the number of fine-tuning iterations (Table 4), keeping the number of fine-tuning examples fixed to 100, and the number of fine-tuning examples (Table 5), keeping the number of fine-tuning iterations fixed to 100.

| # iterations | 10 | | 50 | | 100 | | 500 | |
|---|---|---|---|---|---|---|---|---|
| | R-IB (↑) | D-IB (↑) | R-IB (↑) | D-IB (↑) | R-IB (↑) | D-IB (↑) | R-IB (↑) | D-IB (↑) |
| **RNN** | 0.759 (0.020) | 0.631 (0.030) | 0.670 (0.022) | 0.756 (0.025) | 0.632 (0.023) | 0.797 (0.023) | 0.550 (0.027) | 0.868 (0.019) |
| **LSTM** | 0.805 (0.020) | 0.510 (0.037) | 0.576 (0.029) | 0.605 (0.034) | 0.563 (0.030) | 0.610 (0.034) | 0.553 (0.030) | 0.615 (0.034) |
| **Transformer** | 0.775 (0.022) | 0.585 (0.032) | 0.712 (0.024) | 0.619 (0.033) | 0.703 (0.025) | 0.624 (0.033) | 0.714 (0.024) | 0.629 (0.033) |
| **Mamba** | 0.775 (0.019) | 0.730 (0.025) | 0.698 (0.021) | 0.707 (0.028) | 0.682 (0.021) | 0.728 (0.027) | 0.683 (0.021) | 0.710 (0.028) |
| **Mamba-2** | 0.766 (0.022) | 0.663 (0.031) | 0.653 (0.022) | 0.693 (0.029) | 0.653 (0.022) | 0.694 (0.029) | 0.673 (0.022) | 0.692 (0.029) |

*Table 4.* Results for ablating the number of iterations of fine-tuning.

## F. Additional Transfer Results

Table 6 shows the full transfer learning results described in Section 4.

| # examples | 10 | | 50 | | 100 | | 500 | |
|---|---|---|---|---|---|---|---|---|
| | R-IB (↑) | D-IB (↑) | R-IB (↑) | D-IB (↑) | R-IB (↑) | D-IB (↑) | R-IB (↑) | D-IB (↑) |
| **RNN** | 0.815 (0.024) | 0.384 (0.038) | 0.701 (0.024) | 0.695 (0.033) | 0.632 (0.023) | 0.797 (0.023) | 0.475 (0.022) | 0.930 (0.010) |
| **LSTM** | 0.750 (0.030) | 0.374 (0.039) | 0.625 (0.028) | 0.543 (0.037) | 0.563 (0.030) | 0.610 (0.034) | 0.483 (0.021) | 0.832 (0.020) |
| **Transformer** | 0.862 (0.019) | 0.363 (0.038) | 0.721 (0.022) | 0.610 (0.032) | 0.703 (0.025) | 0.624 (0.033) | 0.578 (0.021) | 0.853 (0.018) |
| **Mamba** | 0.821 (0.020) | 0.456 (0.039) | 0.666 (0.023) | 0.763 (0.026) | 0.682 (0.021) | 0.728 (0.027) | 0.654 (0.018) | 0.864 (0.014) |
| **Mamba-2** | 0.848 (0.018) | 0.453 (0.039) | 0.704 (0.023) | 0.684 (0.030) | 0.653 (0.022) | 0.694 (0.029) | 0.644 (0.024) | 0.886 (0.015) |

*Table 5.* Results for ablating the number of examples used for fine-tuning.

| | | **Majority Tiles** | | **Board Balance** | | **Edge Balance** | |
|---|---|---|---|---|---|---|---|
| | Pretraining | NLL (↓) | ACC (↑) | NLL (↓) | ACC (↑) | NLL (↓) | ACC (↑) |
| **RNN** | Untrained | 0.492 (0.004) | 0.755 (0.003) | 0.405 (0.005) | 0.806 (0.003) | 0.462 (0.002) | 0.816 (0.002) |
| | NTP trained | 0.431 (0.004) | 0.792 (0.002) | 0.302 (0.004) | 0.856 (0.002) | 0.080 (0.002) | 0.964 (0.001) |
| **LSTM** | Untrained | 0.436 (0.004) | 0.786 (0.003) | 0.305 (0.004) | 0.864 (0.002) | 0.105 (0.002) | 0.953 (0.001) |
| | NTP trained | 0.232 (0.004) | 0.901 (0.002) | 0.164 (0.003) | 0.927 (0.001) | 0.041 (0.002) | 0.982 (0.001) |
| **Transformer** | Untrained | 0.497 (0.004) | 0.754 (0.003) | 0.340 (0.005) | 0.855 (0.002) | 0.075 (0.002) | 0.967 (0.001) |
| | NTP trained | 0.100 (0.002) | 0.956 (0.001) | 0.086 (0.002) | 0.965 (0.001) | 0.013 (0.001) | 0.996 (0.000) |
| **Mamba** | Untrained | 0.377 (0.004) | 0.816 (0.002) | 0.246 (0.004) | 0.888 (0.002) | 0.099 (0.002) | 0.952 (0.001) |
| | NTP trained | 0.149 (0.003) | 0.937 (0.002) | 0.158 (0.003) | 0.931 (0.002) | 0.027 (0.001) | 0.989 (0.001) |
| **Mamba-2** | Untrained | 0.379 (0.004) | 0.821 (0.002) | 0.258 (0.004) | 0.891 (0.002) | 0.068 (0.001) | 0.969 (0.001) |
| | NTP trained | 0.069 (0.002) | 0.970 (0.001) | 0.059 (0.002) | 0.976 (0.001) | 0.012 (0.002) | 0.995 (0.001) |
| **IB Correlation** | — | 0.462 | 0.477 | 0.610 | 0.653 | 0.970 | 0.960 |

*Table 6.* Results showing transfer performance across new functions of state. "NLL" represents negative log-likelihood (lower is better), and "ACC" represents accuracy (higher is better). "IB Correlation" measures the (unsigned) correlation between each column of results to the ratios of the inductive bias metrics in Table 2, $\frac{\text{R-IB}}{\text{D-IB}}$. Transfer learning results are correlated to the inductive bias metrics; models with low inductive bias perform worse at transfer.

## G. Next Token Performance

Table 7 shows results for the next-token test (Toshniwal et al., 2022; Li et al., 2023) for the pre-trained models on the lattice and Othello models. It measures the share of top model predictions that are true for the underlying state. All models learn good next token predictions that appear to obey state.

Table 8 shows results for physics. Across 200 held-out trajectories, we autoregressively generate the model's predicted trajectory given the first 50 steps. Then, we compute the MSE of the predicted trajectory, $1, 5, 10$ steps from the $50^{th}$ step. We include the MSE of a baseline that always predicts the most recent timestep.

## H. What are models using to extrapolate?

Here we describe how we compute the decomposition of D-IB into D-IB$_{q=}$ and D-IB$_{q\neq}$. For lattice, we coarsen the state-space by defining a mapping from the ground-truth state-space (of size $N = 5$) to pseudo-state-space of size 3. The mapping is defined as $\{1\} \to 1', \{2, \ldots, N-1\} \to 2', \{N\} \to 3'$.

| | Lattice | Othello |
|---|---|---|
| **RNN** | 1.00 | 0.992 |
| **LSTM** | 1.00 | 0.996 |
| **Transformer** | 1.00 | 0.999 |
| **Mamba** | 1.00 | 0.999 |
| **Mamba-2** | 1.00 | 0.999 |

*Table 7.* Results for the next token test (Toshniwal et al., 2022; Li et al., 2023) for models pre-trained on next-token prediction.

| # steps out | 1 | 5 | 100 |
|---|---|---|---|
| **Per-orbit mean** | $(7.53 \pm 0.59) \cdot 10^{-2}$ | $(5.53 \pm 0.58) \cdot 10^{-2}$ | $(1.39 \pm 0.08) \cdot 10^{-1}$ |
| **Previous position** | $(1.16 \pm 0.21) \cdot 10^{-4}$ | $(1.37 \pm 0.38) \cdot 10^{-4}$ | $(4.04 \pm 0.47) \cdot 10^{-2}$ |
| **Transformer** | $\mathbf{(1.90 \pm 0.45) \cdot 10^{-8}}$ | $\mathbf{(1.56 \pm 0.45) \cdot 10^{-8}}$ | $\mathbf{(3.74 \pm 3.37) \cdot 10^{-5}}$ |

*Table 8.* Orbit trajectory prediction performance (MSE) for models pre-trained on next-token prediction. Each column shows prediction accuracy when forecasting planetary positions 1, 5, or 100 time steps ahead from position 500 in the sequence. We compare the transformer model to two simple baselines (one that always predicts a planet's position at the previous timestep, and another that uses the per-orbit mean). All results are evaluated on held-out test trajectories.

| | **Lattice** | | **Othello** | |
|---|---|---|---|---|
| | D-IB$_{q=}$ | D-IB$_{q\neq}$ | D-IB$_{q=}$ | D-IB$_{q\neq}$ |
| **RNN** | 0.740 (0.042) | 0.844 (0.034) | 0.521 (0.031) | 0.798 (0.023) |
| **LSTM** | 0.873 (0.051) | 0.952 (0.034) | 0.519 (0.035) | 0.610 (0.034) |
| **Transformer** | 0.626 (0.037) | 0.710 (0.037) | 0.458 (0.033) | 0.625 (0.033) |
| **Mamba** | 0.764 (0.040) | 0.933 (0.035) | 0.485 (0.030) | 0.729 (0.027) |
| **Mamba-2** | 0.778 (0.042) | 0.920 (0.033) | 0.553 (0.032) | 0.694 (0.029) |

*Table 9.* Metrics for assessing whether a model's inductive bias is toward its legal next-token partition. Low values of D-IB$_{q=}$ and high values of D-IB$_{q\neq}$ suggest that failures to differentiate state are driven by the models having an inductive bias toward the legal next-token partition.

For Othello, we coarsen the state-space by defining a mapping from board state to the set of legal next moves possible from the state. Notice that this mapping is many-to-one: as the pair of boards in Figure 4 demonstrate, there can be many boards that share the same set of legal next moves.

Then, we measure the expected extrapolative predictability of a random pair of sequences that have different states but the same pseudo-state (D-IB$_{q=}$) and a random pair of sequences that have both different states and also different pseudo-states (D-IB$_{q\neq}$), as defined in Section 4.

The results are shown in Table 9. Note that across all models, D-IB$_{q=}$ is smaller than D-IB$_{q\neq}$. In other words, among sequences with different states, extrapolations on sequences that share the same legal next tokens are more predictable from each other than those on sequences that do not.

# I. Additional Figures

Figure 9 shows an overview of the inductive bias probe. Figure 10 illustrates the inductive bias probe in the special case where the given world model has a finite state space.

## Inductive bias probe

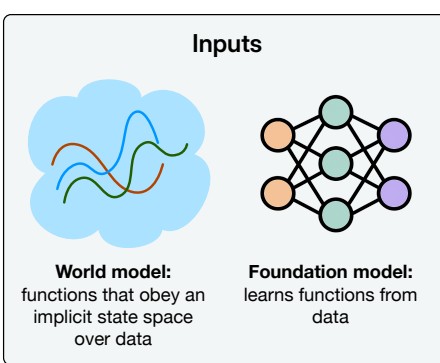

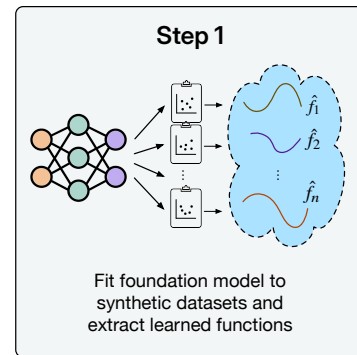

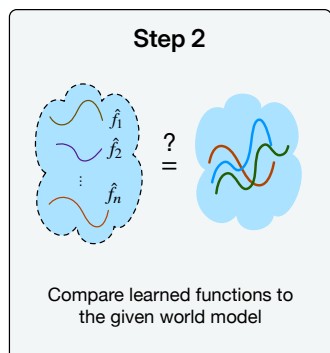

*Figure 9.* An inductive bias probe measures whether a foundation model has an inductive bias toward a given world model. The probe involves repeatedly fitting a foundation model to small, synthetic datasets and comparing the functions it learns to the functions in the given world model.

## Example: Finite state space

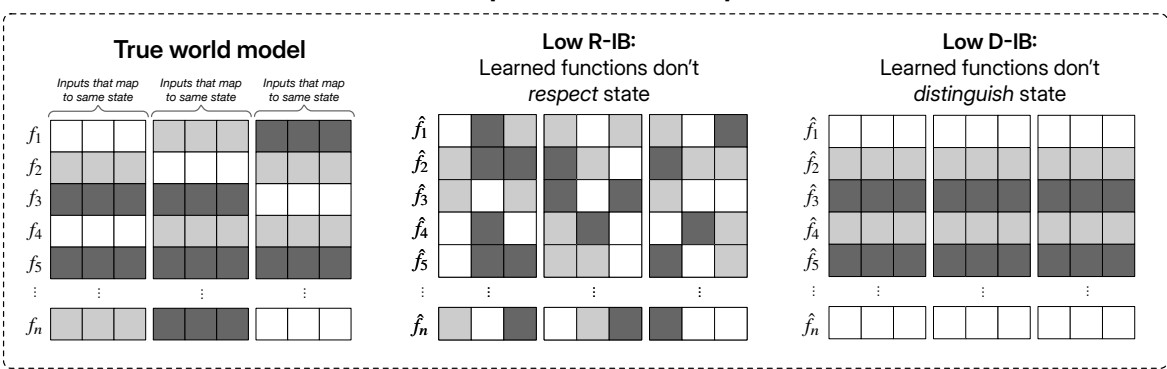

*Figure 10.* An illustration of the inductive bias probe when the given world model has a finite state space. Each row represents a function and each column represents an input $x_i$, with inputs belonging to the same state grouped together. The shading illustrates each function's value at the corresponding input. A foundation model has low R-IB (middle) if it learns functions that divide states, while a foundation model has low D-IB (right) if it learns function that merge states.

