# OpenReview forum: "What Has a Foundation Model Found? Using Inductive Bias to Probe for World Models"
_ICML.cc/2025/Conference — ICML 2025 poster_

### Official Review · Reviewer_86Gp · 2025-03-11

**Overall Recommendation:** 3

**Summary:**

This paper proposes methods for evaluating if the predictions made by foundation models following fine-tuning on new tasks are compatible with a reference world model.  The authors introduce two metrics; inductive bias towards respecting state and inductive bias towards distinguishing state, which aim to measure how well the foundation model's extrapolations align with a reference world model. They demonstrate their approach in the domains of classical mechanics (orbital bodies), lattice problems, and the game Othello, and give evidence that foundation models trained on next-token prediction fail to transfer to new tasks in a way that is consistent with the underlying world model, but instead rely on task-specific heuristics..


## update after rebuttal

Following the authors rebuttal, and proposed clarifications of the paper, I am now learning towards acceptance.

**Claims And Evidence:**

The authors do present "a framework for testing foundation models by analyzing how they adapt to synthetic tasks that share mechanisms with their training domain". They give quite convincing evidence that the models they evaluate are not generalizing in a way that respects the underlying statespace.

One of the key claims—that foundation models learn heuristics rather than fundamental world models—seems to come mostly from symbolic regression on the predictions of a model, rather than applying the metrics that are the main results of the paper. It is not clear how robust this claim is from this single, and quite small, experiment.

**Essential References Not Discussed:**

NA

**Experimental Designs Or Analyses:**

Yes, pre-training and fine-tuning settings are well-described. Baseline models (RNNs, LSTMs, Transformers, Mamba) are compared, and the novel metrics (R-IB, D-IB) are computed over multiple training seeds. Symbolic regression is used to validate whether models recover correct physical laws, which I am less certain about. The experiment is fairly small, and its not immediately clear how robust the result is, and if its really coming from the fact that the models are learning the wrong physical laws.

**Methods And Evaluation Criteria:**

The metric R-IB seems well motivated to me, though I struggled to understand the metric D-IB (see questions below)

**Other Comments Or Suggestions:**

I will increase my score if the Framework section is sufficiently improved, and if the language is toned down a bit re: the results showing that the foundation model has learned a world model (rather than, to what degree its predictions are consistent with a given world model).

**Other Strengths And Weaknesses:**

Overall I think the proposed method is compelling, though Im not sure it quite does what the authors claim and establishes that a world model has been learned. The main weakness to my mind is the formalism section, which I found very unclear and had to read multiple times before realising (I think) what the authors are proposing. See notes below.

**Questions For Authors:**

Question 1

My current understanding of the formalism the authors propose is this, and my first question is if this accurately captures what they are proposing.

1. The authors define consistency in the following way. Assume a foundation model makes predictions using a world model prediction = f(state = WM(input)). Assume the foundation model predicts in this way, and that the WM is re-used across tasks (e.g. when we fine-tune the foundation model on some new task). Then, if two inputs map to the same state in the world model, then they must give the same predictions across all tasks, where the task is to prediction some function that depends only on the state (e.g. not any additional information in the input that is not captured in the state, such as how the state was reached).
2. They then invert this relationship, saying that if two inputs are treated the same across all tasks, then they are compatible with being the same states in a world model. This does not imply they are the same states in the world model, or even that there is a world model, just that the foundation models predictions are compatible with this being true.
3. Then, if we have a reference world model (such as the statespace of a dynamical system, or the board state of a game), we can map inputs (e.g. trajectories) to states on the `correct’ world model. Then, for the foundational model to be compatible with this correct world model, the predictions must be the same for all inputs x that correspond to the same state on the correct world model. If this is not true, then the world model implicit in the foundation model (if there is one) is to some extend incompatible with the correct or reference world model.

If this is correct, then the introduction of the framework could have been a lot clearer and easier to understand. It took me multiple reads to get to this understanding, that I am still not confident on. But (if the above is accurate), the core ideas are not very complex, and will be better motivated by a simplified introduction.

Question 2

It could be that the foundation model has learned the correct world model but its predictions cannot be described by prediction = f(state = WM(input)). For example, if prediction = f(state = WM(input), input), e.g. the prediction is not just based on the world state but some spurious information from the input (such as having a bias towards certain predictions depending on the input length). This would give a false negative (there is an accurate world model, but a poor process of drawing inferences from that model). The model prediction = f(state = WM(input), input) even makes sense if assume the world model is encoded in the intermediate layers of the transformer, and the action of the residual stream. Does your result get around this in some way? Its fine if not, but it would be nice to A) have all the assumptions on how the foundation model generates predictions in a single paragraph, B) some examples of possible ways foundation models could generate predictions (as described here) that violate the assumptions (including false positives, where there is no world model, but the foundation model passes the eval).

Question 3

the motivation and intuition for D-IB is not clear when it is introduced. When you say about D-IB “ If a learning algorithm’s effective foundation is based on the world model, it should now extrapolate in predictable ways between inputs values in different state”, what precisely do you mean here by "in predictable ways"? What should we expect D-IB to be if there is a world model, and why? Are you saying something like, if the predictions are based on the (true) world model, then the correlation between predictions that map to the same state should be invariant between different synthetic tasks? Or that the correlation should be invariant when we vary the inputs x_i, x_j but they still map to fixed (but different) states?

Question 4


You talk about foundation models learning the correct underlying physical laws of a system, but focus purely primarily on the state space. Physical laws involve two kinds of object; kinematics (states) and dynamics, and the dynamics are typically the most important thing as they allow for future states to be predicted. For example in RL, world models capture both of these fundamental objects. In the newtonian mechanics example, the most interesting part in my opinion was the symbolic regression on the actual predictions, precisely because you are seeing if the model is learning the correct dynamics or not. Can you explain what R-IB and D-IB are telling you here, in simple terms? And am I right in thinking that the main claim, that the models are learning task-specific heuristics, comes from these symbolic regressions? If so, it feels like this should be a larger experiment, with a lot more details on experimental design.

**Relation To Broader Scientific Literature:**

There is a good discussion of related work as far as I can see.

**Theoretical Claims:**

NA

---

> ### Author Rebuttal · Authors · 2025-03-31
>
> Thank you for your careful and insightful review.
>
> > _I will increase my score if the Framework section is sufficiently improved, and if the language is toned down a bit re: the results showing that the foundation model has learned a world model (rather than, to what degree its predictions are consistent with a given world model)._
>
> We really like the way you've described the framework and we've modified our paper to match (see below).
>
> We also agree with your language change: it's overstating to say “has learned a world model”; given our test, it's much more accurate to say “consistent with a given world model”, which is the language we'll use going forward.
>
> Your comments were very constructive and have improved the paper. **We believe we've addressed your comments below; if we haven't, please let us know if you have any further questions.**
>
>
> > _Question 1: ...My first question is if this accurately captures what they are proposing..._
>
> We've rewritten the framework to incorporate your suggestions. We don't have space to include the full section here but the outline is:
>
> - This paper is about measuring the degree to which a foundation model's predictions are consistent with a given world model.
> - Note a mechanistic interpretation is difficult to measure.
> - Instead we examine model predictions on synthetic tasks that obey the world model
> - Consistency requires two properties:
>   1. R-IB: When two inputs map to the same state in the world model, do they have the same predictions for each task?
>   2. D-IB: When two inputs have the same predictions for each task, do they belong to the same state?
> - R-IB and D-IB are different perspectives on consistency, analogous to type-I/type-II errors.
>
> > _Question 2: It could be that the foundation model has learned the correct world model but its predictions cannot be described by prediction = f(state = WM(input))_
>
> Great question. This is why we have _two_ metrics:
> - If a model predicts f(state, input) != f(state, input') for same-state inputs, R-IB is low
> - If a model predicts f(state, input) = f(state', input') for different-state inputs, D-IB is low
>
> These metrics can be used independently (e.g. focusing only on R-IB). But there are many applications where we’d care if a model has extraneous information. For example, if a model carries extraneous information its predictions on small data may be subject to spurious correlations. Empirically we find that low D-IB scores suggest that models base their predictions not on true state but on next-token partitions (see Figure 2 and Table 8), hurting small-data performance.
>
> > _Question 3: When you say about D-IB “If a learning algorithm’s effective foundation is based on the world model, it should now extrapolate in predictable ways between inputs values in different state”, what precisely do you mean?_
>
> You've helped us identify an unfortunate typo. The sentence should read "...it should **not** extrapolate in predictable ways...". So if the model learns according to not only the effective foundation but other inputs, D-IB is lower. We've corrected this.
>
> > _Question 4: Physical laws involve two kinds of object; kinematics (states) and dynamics..._
>
> We want to make sure we're interpreting your question correctly:
> - kinematics: mapping from inputs X to state space (i.e. given by phi)
> - dynamics: how the state spaces evolve.
>
> We agree the distinction is interesting. Our framework only addresses kinematics (e.g. state, not the laws that may govern dynamics on top of state). Therefore R-IB and D-IB tell us about kinematics. But the two are related; knowing kinematics often allows recovering dynamics from enough sequences.
>
> We focus on states because even in physics there's value in studying kinematics independently from dynamics. Although orbital trajectories (dynamics) are fixed, many interesting functions depend on state (e.g. energy, angular momentum). Addressing dynamics explicitly is valuable future work.
>
> > _[Does the result that] models are learning task-specific heuristics [come only] from these symbolic regressions?_
>
> Symbolic regressions are one tool for showing that models learn task-specific heuristics (we've added more experimental details in the appendix), and it's a common method for validating physics models [1, 2]. But it's not the only tool; our appendix has additional experiments with other methods (Table 8) in non-physics domains. We've moved this to the main text.
>
> We've also added added new physics experiments to Table 8:
> - D-IB for distinct states with same next tokens: 0.709
> - D-IB for distinct states with different next tokens: 0.764
>
> So poor D-IB can be partially attributed to models basing their predictions not on state but on next-token partitions. Figure 2 illustrates this point for Othello.
>
> [1]  Udrescu et al. "AI Feynman: a Physics-Inspired Method for Symbolic Regression." (2020).
> [2] Cranmer et al. "Discovering Symbolic Models from Deep Learning with Inductive Biases." (2020).

---

> > ### Comment · Reviewer_86Gp · 2025-04-02
> >
> > Thank you for your reply, which has cleared up a lot of my confusion and if implemented would improve the clarity of the paper greatly. I have raised my score.

---

### Official Review · Reviewer_kBZN · 2025-03-13

**Overall Recommendation:** 4

**Summary:**

In this paper, the authors focus on the problem of understanding the generalization capabilities of foundational models. For example, can a foundational model truly develop an inductive bias towards Newtonian mechanics or the rules of a board game? The authors test this question, developing a framework for testing world models in addition to proposing metrics such as Inductive Bias towards Respecting State (R-IB) and inductive bias towards distinguishing state (D-IB). The test hypothesis in a number of domains - specifically orbital mechanics, the board game Othello, and Lattices. They find that while common world models such as transformers can excel at specific tasks, they fail to learn inductive biases and rely on data-specific, piecemeal heuristics.

## update after rebuttal
In line with the other reviewers' comments, I keep my assessment of accept.

**Claims And Evidence:**

Yes. The authors clearly show quantitatively and qualitatively that world models struggle with learning general principles from data.

**Essential References Not Discussed:**

No.

**Experimental Designs Or Analyses:**

Experimental designs are sound even if the datasets used are simplistic and metrics abstract.

**Methods And Evaluation Criteria:**

Yes. Even though the domains used for evaluation are very simple (two-world body physics, 2d-board game), they allow for a deeper understanding why foundational models fail to generalize. The qualitative failure cases (such as the Figure 1 and 2) make the strongest cases for the paper. The quantitative metrics such as R-IB and D-IB also support the argument even if they are somewhat abstracted away from the domains.

**Other Comments Or Suggestions:**

None.

**Other Strengths And Weaknesses:**

The proposed metrics (R-IB and D-IB), while needed for a comprehensive evaluation, are somewhat abstract and sometimes difficult to understand in the context of the various domains (Orbital physics, Othello). It would be helpful to ground the concepts for these metrics with domain-specific examples when describing them in section 2.

**Questions For Authors:**

I have no questions for the authors.

**Relation To Broader Scientific Literature:**

This paper contests the idea that foundational models are able to learn inductive biases in various domains and shows specifically how and why they fail. One of the domains (Othello) has been used in prior literature.

**Theoretical Claims:**

The paper contains no proofs.

---

> ### Author Rebuttal · Authors · 2025-03-31
>
> Thank you for your positive review of our paper. We're glad that you appreciated our paper and findings, and that our results gave you "a deeper understanding [of] why foundation models fail to generalize".
>
> > _The proposed metrics (R-IB and D-IB), while needed for a comprehensive evaluation, are somewhat abstract and sometimes difficult to understand in the context of the various domains (Orbital physics, Othello). It would be helpful to ground the concepts for these metrics with domain-specific examples when describing them in section 2._
>
> This is a great point. We've revised our paper to make the framework clearer, and we've added more domain-specific examples.
>
> Specifically:
> - A foundation model's predictions are consistent if two properties hold:
>   1. When two inputs map to the same state in the world model, do they have the same predictions for each synthetic task? This is measured by R-IB.
>   2. When two inputs have the same predictions for each synthetic task, do they belong to the same state? This is measured by D-IB.
> - So R-IB and D-IB are different perspectives on measuring consistency, analogous to type-I and type-II errors in classification.
>
> Here's an implementation example for the lattice problem:
> - X is a sequence of directions ("L", "R", "stay")
> - 5 states (1-5)
> - phi(X) maps movement in X starting at state 1
>   - So phi("R", "R", "L")=2 (two states right, one state left)
> - We create random synthetic datasets (X, Y) such that phi(x) = phi(x') implies y = y' for each (x,y), (x',y') pair.
> - We fit a foundation model (e.g. by fine-tuning) on a very small subset of each dataset and make predictions for the held-out data points.
> - R-IB: do two held-out points with the same state always have the same predictions across synthetic datasets?
> - D-IB: do two held-out points with different states have unpredictable extrapolations across synthetic datasets?

---

### Official Review · Reviewer_Genp · 2025-03-13

**Overall Recommendation:** 3

**Summary:**

This paper investigates whether foundation models trained via next-token prediction learn "world models". The paper uses synthetic tasks to measure whether such models are able to generalize from their training tasks to other tasks drawn from the same distribution. The evaluation is performed using two metrics to quantify how much the model learns inductive biases that _respect_ and _distinguish_ ground-truth states, according to an expert-defined history-to-state mapping. The paper includes experiments in predicting orbital mechanics, Othello board states, and discrete position on a number line, and presents evidence that despite good performance at next-token prediction, models appear not to use the expert-defined state representations.

**Claims And Evidence:**

Claims:

1. "Framework for testing foundation models by analyzing how they adapt to synthetic tasks that share mechanisms with their underlying training domain."

    - Supported. The framework itself appears sound. The orbital mechanics dataset samples random initial conditions (masses, positions, and relative velocities), and shows that the model can predict the trajectories without correctly predicting force vectors. I believe this demonstrates that the framework can be applied for the intended purpose.
- Measuring the model's ability to predict sequences it was not trained on feels like an appropriate way to test generalization, and true generalization [seems to require](https://arxiv.org/abs/2402.10877) a (causal) world model. That said, I believe the linked paper makes that claim with respect to out-of-distribution data, whereas this paper assumes the generalization is still in-distribution.

2. "Metrics that measure a model's inductive bias toward known models of reality."

    - I had trouble evaluating this. I found the inductive bias definitions for respecting and distinguishing state to be a bit hard to follow. In Section 2: Implementation, a toy example would have been helpful. My impression is that if the approach fails, it could be that the predictor model is underpowered, rather than due to a lack of world model.
    - The defined metrics appear to rely heavily on whether two inputs share the same underlying state. It would be helpful to quantify what fraction of the dataset contains different inputs from the same state. I'm concerned that if there aren't many such examples, the metrics might behave poorly. For example, in Othello there are a combinatoric number of states, and after a certain number of steps, it is highly unlikely that the model will see the same state twice for two different trajectories.

3. "Models can excel at their training tasks yet fail to develop inductive biases toward the true mechanisms when being adapted to new tasks."

    - Mostly supported: I already mentioned the orbital mechanics experiments, and the Othello experiments also showed that the model struggles to predict the correct board state but successfully predicts the set of next legal moves.

**Essential References Not Discussed:**

N/A

**Experimental Designs Or Analyses:**

See above.

**Methods And Evaluation Criteria:**

The definition of a "world model" as a mapping from histories to a particular ground-truth state representation feels problematic for answering the ultimate question of whether foundation models contain world models. There are many valid and useful state representations, and if the foundation model fails to match a particular expert-defined representation, it does not mean the foundation model lacks a world model altogether.

For example, [this project](https://www.neelnanda.io/mechanistic-interpretability/othello) showed that when evaluating whether Othello-GPT had a linear representation of the board state, it matters whether the representation is of the form "this cell is black" vs. "this cell has my color", since the model plays games as both colors.

**Other Comments Or Suggestions:**

N/A

**Other Strengths And Weaknesses:**

I wonder how useful this method can be when there multiple acceptable state representations. It would be helpful for the paper to analyze, for example, what happens in Othello under the two representations described above.

**Questions For Authors:**

Can you provide a more intuitive explanation of the inductive bias definitions?

Can you provide a toy example to go along with the Section 2: Implementation?


=====================

POST-REBUTTAL UPDATE

=====================

I found the comments by reviewer 86Gp and the subsequent response very helpful in understanding the paper's main claim. I agree that explanation could have been much clearer, and I'm happy the authors are taking steps to improve it.

In the rebuttal "Lattice problem" example, I am still unclear on what exactly $y$ is in the pair $(x,y)$. I suppose it's some predicted quantity that depends on the encoded state $\phi(x)$? I would encourage the authors to make this example even more concrete by saying what specific thing is being predicted, how it depends on the state, and maybe even provide some example predictions from which we can see what the D-IB and R-IB metrics would report.

I thank the authors for providing results for the two different Othello representations. Unfortunately I'm not sure what you mean by "poor IB metrics", and I don't understand what the takeaway is here. In regards to my original question of whether the method is sensitive to the choice of representation, it sounds like the answer is no. But then why are there two different correlation numbers for the different distance metrics? I would encourage the authors to take a bit more time to explain this.

Overall, I think the weakest aspect of the paper is the clarity of the presentation, and I urge the authors to do whatever they can to clarify the points that the other reviewers and I were confused about.

**Relation To Broader Scientific Literature:**

It is an extremely important question whether foundation models learn world models. Foundation models are the dominant paradigm in AI right now, and much of our confidence in them is predicated on the assumption that their performance at sequence prediction implies that they have learned a world model.

This paper doesn't address the question fully. Instead it addresses the question of whether a particular foundation model has learned a particular world model. A satisfying answer to this narrower question would still be a useful step towards answering the broader question.

**Theoretical Claims:**

N/A

---

> ### Author Rebuttal · Authors · 2025-03-31
>
> Thank you for your positive review of our paper. We respond to your comments and describe new results below; to summarize the new results, we've added:
> - New experiments showing robustness to predictor in R-IB and D-IB calculations
> - A more intuitive explanation of our framework
> - Clarification and new analysis of multiple representations in Othello
>
> **We hope our comments have addressed your concerns. If not, please let us know if you have any more questions we can address in the follow-up.**
>
> > _Can you provide a more intuitive explanation of the inductive bias definitions? Can you provide a toy example to go along with the Section 2: Implementation?_
>
> We've revised our paper to make this more intuitive. Broadly:
> - A foundation model's predictions are consistent if two properties hold:
>   1. When two inputs map to the same state in the world model, do they have the same predictions for each synthetic task? This is measured by R-IB.
>   2. When two inputs have the same predictions for each synthetic task, do they belong to the same state? This is measured by D-IB.
> - So R-IB and D-IB are different perspectives on measuring consistency, analogous to type-I and type-II errors in classification.
>
> Here's an implementation example for the lattice problem:
> - X is a sequence of directions ("L", "R", "stay")
> - 5 states (1-5)
> - phi(X) maps movement in X starting at state 1
>   - So phi("R", "R", "L")=2 (two states right, one state left)
> - We create random synthetic datasets (X, Y) such that phi(x) = phi(x') implies y = y' for each (x,y), (x',y') pair.
> - We fit a foundation model (e.g. by fine-tuning) on a very small subset of each dataset and make predictions for the held-out data points.
> - R-IB: do two held-out points with the same state always have the same predictions across synthetic datasets?
> - D-IB: do two held-out points with different states have unpredictable extrapolations across synthetic datasets?
>
> > _My impression is that if the approach fails, it could be that the predictor model is underpowered, rather than due to a lack of world model_
>
> We can think of two ways to interpret this question: one is about the predictor that's part of the foundation model; the other is about the predictor used to calculate R-IB and D-IB.
>
> For the first interpretation: We're interested in understanding a foundation model's extrapolative properties in small data regimes (e.g. few shot learning) since this is how they're typically used. Thus, our metrics measure consistency with the world model, not predictor accuracy, so they wouldn't penalize an underpowered predictor.
>
> For the second interpretation: R-IB and D-IB are based on a predictor that only uses a single binary feature. Therefore the Bayes optimal solution is nonparametrically identifiable. [This table varies the number of data points used to form R-IB and D-IB](https://imgur.com/a/ppaDzrx); empirically we find little sensitivity, because the predictor converges to the optimal solution quickly.
>
> > _I wonder how useful this method can be when there [are] multiple acceptable state representations. It would be helpful for the paper to analyze... what happens in Othello under the two representations described above._
>
> This is an important question. Our framework depends on the state mapping, not how it's represented. The only important quantities for D-IB and R-IB are whether two inputs have the same or different state. This means the **metrics aren't sensitive to different ways state can be represented**, so the two Othello representations have identical metrics. In comparison, sensitivity to representation makes other metrics (e.g. probes) more fragile.
>
> However we could still use different representations to help _analyze_ poor IB metrics. D-IB measures how similar the predictions for input pairs with distinct states are. We consider two distance metrics for distinct-state pairs: standard ("cell is black") vs. relative ("cell has my color") Hamming distances. We find:
> - Corr(D-IB, standard distance) = 0.094
> - Corr(D-IB, relative distance) = 0.001
>
> Interestingly this indicates states with similar standard representations are more easily confused.
>
> > _I believe... this paper assumes the generalization is still in-distribution._
>
> This is a good point. It's crucial to also test out-of-distribution (OOD). While our submission was unclear, we do test on OOD sequences -- the only restriction on synthetic tasks is that they be consistent with the world model, not the original sampling distribution. We've changed the writing to make this clear.
>
> > _It is highly unlikely that the model will see the same state twice for two different trajectories [in Othello]._
>
> This is absolutely right. Going back to your earlier point, it shows the importance of testing distribution shifts. In practice, we sample inputs in Othello so that we have enough same-state examples (3,889) to make accurate estimates. Appendix B has more information but we'll highlight it in the main text.

---

### Decision · Program_Chairs · 2025-05-01

**Decision:**

Accept (poster)

**Comment:**

The paper proposes a novel framework and metrics to test if foundation models learn underlying world models using synthetic tasks. Reviewers appreciate the significant and timely research question, the valuable novel framework and metrics, and the use of well-chosen synthetic tasks. However, initial concerns were raised regarding the clarity of the metrics/framework and the robustness of some analyses. Overall, the paper is recommended for acceptance, as the authors' rebuttal addressed concerns, confirming the work's valuable contribution.